# Sentinel Lymph Node Biopsy in Prostate Cancer: An Overview of Diagnostic Performance, Oncological Outcomes, Safety, and Feasibility

**DOI:** 10.3390/diagnostics13152543

**Published:** 2023-07-31

**Authors:** Giulio Rossin, Federico Zorzi, Pedro De Pablos-Rodríguez, Arianna Biasatti, Josè Marenco, Luca Ongaro, Alessandro Perotti, Gabriele Tulone, Fabio Traunero, Andrea Piasentin, Alvaro Gomez-Ferrer, Alessandro Zucchi, Carlo Trombetta, Alchiede Simonato, José Rubio-Briones, Riccardo Bartoletti, Miguel Ramírez-Backhaus, Francesco Claps

**Affiliations:** 1Urological Clinic, Department of Medicine, Surgery and Health Sciences, University of Trieste, Strada di Fiume, 447, 34149 Trieste, Italy; giulio.rossin93@gmail.com (G.R.); federicozorzi94@gmail.com (F.Z.); arianna.biasatti@gmail.com (A.B.); ongarluc@gmail.com (L.O.); fabio.tra92@gmail.com (F.T.); andrea.piasentin23@gmail.com (A.P.); trombcar@units.it (C.T.); 2Department of Urology, Valencian Oncology Institute Foundation, FIVO, 46009 Valencia, Spain; pablos.rodriguez92@gmail.com (P.D.P.-R.); jlmarencoj@gmail.com (J.M.); alvarogomezferrer@gmail.com (A.G.-F.); ramirezfivo@gmail.com (M.R.-B.); 3Department of Translational Research and New Technologies, University of Pisa, 56126 Pisa, Italy; perotti.alessandro@yahoo.it (A.P.); zucchi.urologia@gmail.com (A.Z.); riccardo.bartoletti@unipi.it (R.B.); 4Urology Clinic, Department of Surgical, Oncological and Stomatological Sciences, University of Palermo, 90133 Palermo, Italy; gabriele.tulone@gmail.com (G.T.); alchiede@gmail.com (A.S.); 5Clínica de Urología, Hospital VITHAS 9 de Octubre, 46015 Valencia, Spain; jrubio@clinicadoctorrubio.es

**Keywords:** prostate cancer, extended pelvic lymph node dissection, nodal staging, sentinel lymph node biopsy, sentinel lymph node, indocyanine green, fluorescence, radio-guided surgery

## Abstract

Sentinel node biopsy (SNB) for prostate cancer (PCa) represents an innovative technique aimed at improving nodal staging accuracy. The routinary adoption of this procedure in patients undergoing radical prostatectomy (RP) might be crucial to identify candidates who could effectively benefit from extensive pelvic lymph nodal dissection (ePLND). Despite some promising results, SNB for PCa is still considered experimental due to the lack of solid evidence and procedural standardization. In this regard, our narrative review aimed to analyze the most recent literature in this field, providing an overview of both the diagnostic accuracy measures and the oncological outcomes of SNB.

## 1. Introduction

Prostate cancer (PCa) represents the most frequently diagnosed malignancy and the second leading cause of cancer death in males, with 268,490 new cases and 34,500 estimated deaths in the United States in 2022 [1,2]. PCa lymph node metastases (LNMs) are reported in almost 15% of patients undergoing radical prostatectomy (RP) and extended pelvic lymph node dissection (ePLND) [3]. According to the European Association of Urology (EAU) guidelines, ePLND should always be performed in high-risk PCa cases as well as in intermediate-risk patients at significant risk of nodal spread, as defined by using validated nomograms [4]. An extended dissection template should include external iliac nodes, the nodes within the obturator fossa located cranially and caudally to the obturator nerve, and those located medially and laterally to the internal iliac artery [5]. Despite these recommendations, there is still some controversy regarding the prognostic significance of ePLND for clinically localized PCa. Moreover, ePLND, as such, is a complex and time-consuming procedure burdened by a non-negligible perioperative and a postoperative complication rate [6].

Preoperative clinical nodal (cN)-staging represents another debated topic in PCa management. Traditional imaging has shown low diagnostic accuracy in cN-status assessment when compared to emerging technologies, such as prostate-specific membrane antigen (PSMA), positron emission tomography (PET), and computed tomography (CT) [7,8]. Although PSMA PET-CT diagnostic capabilities are yet to be fully explored, controversial results were reported in the context of nodal micro-metastasis detection [9,10]. To better stratify the risk of lymph node involvement (LNI), several tools have been explored in a bid to provide a reliable preoperative risk assessment of a patient’s nodal status. In 2017, Gandaglia’s nomogram showed high accuracy in LNMs detection, with only 1.5% of patients with positive nodes being missed using a 7% risk threshold [11]. Despite this, approximately 70% of patients classified at risk of LNI, according to nomograms, still receive an avoidable and unnecessary ePLND, as shown by negative LNs in the final pathology report [11]. Moreover, Barletta et al. demonstrated that less than 2% of patients with clinically localized PCa harbor lymph nodal invasion [12]. Hence, there is still an unmet clinical need to provide a more accurate nodal staging for intermediate- and high-risk patients undergoing RP.

In this regard, PCa sentinel node biopsy (SNB) represents a promising frontier. Historically, the sentinel lymph node (SLN) is defined as the first lymph node station of lymphatic to drain the primary tumor’s site. Hence, SLN dissection could represent a useful tool to screen patients who might benefit from ePLND from those who can safely be spared it. Nowadays, SLN dissection is indeed routinely performed for malignancies such as breast and penile cancer as well as for melanoma as part of a specific and well-defined diagnostic and therapeutic pathway [13,14,15]. Despite the widespread adoption of SNB in the aforementioned settings, SNB has yet to be defined and standardized in PCa. In this context, some drawbacks are represented by the complexity of prostatic pelvic lymphatic drainage, which makes SNB particularly challenging for PCa. In addition, a high procedural heterogeneity is reported in the literature, with a large methodological variability mainly concerning lymphatic tracers, injection sites, and the use of preoperative/intraoperative imaging to detect positive LNs. Thus, SNB procedures have gained increasing attention over the last years as a potential tailored alternative to ePLND. Our narrative review aims to summarize the latest findings on SNB for PCa with a particular focus on procedural aspects, safety, and feasibility of this technique.

## 2. Materials and Methods

A bibliographic search using Medline and EMBASE between January and May 2023 was performed, using as keywords “Prostatic Neoplasms”, “Prostate Cancer”, “Pelvic Lymph Node Dissection”, and “Sentinel Lymph Node”. A total of 140 manuscripts were identified through database search. Among them, 28 studies met the inclusion criteria and were included in the present narrative review. Publications including less than 10 patients as well as those not written in the English language, editorials, comments, case reports, review articles, and conference abstracts were excluded. Moreover, we only considered papers investigating SNB’s role in combination with RP and ePLND [16]. Median values and interquartile ranges (IQRs) for standard diagnostic and oncological outcomes were identified and reported.

## 3. Results

### 3.1. Diagnostic Outcomes

SNB diagnostic performance represents the most investigated theme, with several papers published in recent years and some still ongoing randomized trials on this specific topic. Standard diagnostic outcomes from the included studies are summarized in Table 1. 

Data collected from the included manuscripts accounted for a total of 2586 patients. Regarding SLN tracer selection, a total of 10 papers reported the use of technetium 99m-nanocolloids (99mTc-NC) [18,19,29,33,35,37,38,39,40,41], 6 papers of “free” indocyanine green (ICG) [17,22,28,30,32,44], 3 papers of “hybrid” ICG-99mTC-NC tracer [23,24,31], 4 papers superparamagnetic of iron oxide nanoparticles (SPION) [25,26,27,36], 2 papers of PSMA radiotracer [20,21], 1 paper of technetium 99m-sulfur [34], 1 paper of technetium 99m-phylate [42], and 1 paper of technetium 99m human serum albumine (HAS) [43]. Two groups combined the use of 99mTc-NC and intraoperative ICG within the same cohort [23,24]. Regarding tracer administration, 11 papers reported an intraprostatic injection template [17,18,19,22,24,25,27,29,37,38,43]. Intra-/peri-lesional administration was performed in two reports [18,29]. The mean number of dissected LNs from SNB templates was collected, with an overall mean of 4.9 LNs dissected per SNB template.

A separate “per patient” and “per node” data analysis has been provided by the Authors. Median values and IQRs are reported in Table 2. 

Overall, median “per patient” sensitivity was 92.5% (IQR 82.8–100.0) (24 papers included) and 65.4% “per node” (IQR: 51.5–74.2) (8 papers included), respectively. 

The median negative predictive values (NPV) for “per patient” and “per node” analyses were 97.9% (IQR: 94.2–100.0) (24 papers included) and 98.1% (IQR: 91.0–98.2) (7 papers included), respectively.

Overall, the median false negatives’ (FN) rate, defined as nodal positivity at ePLND final pathology with negative intraoperative SNB, reached as high as 4.6% (IQR: 0–10.2) (20 papers included). The median false positives’ (FP) rate, defined as positive SLNs localized outside the ePLND template, was 0.0% (IQR 0–3.8) (considering 20 papers). 

### 3.2. Oncological Outcomes

While the oncological benefit of ePLND in clinically localized PCa is still a matter of debate, nowadays, even more controversial is the impact of a targeted fluorescence- or radio-guided PLND in such a clinical scenario. The main findings of the included papers are reported in Table 3. 

Regarding oncological outcomes, biochemical recurrence (BCR) was the most investigated [18,19,20,21,23,30,31,33,44]. Due to the relatively recent development of the technique and its still experimental role, only short and mid-term follow-up data are currently available on this topic [24]. Table 4 shows the median values of oncological outcomes described in the included papers. Median rates for PSA persistency and BCR were 17.2% (IQR: 4.6–37.5) and 16.7% (IQR: 14.0–37.5), respectively, at a median follow-up of 16.5 months.

### 3.3. Safety Profile and Complications Rates

While ePLND-related complications in PCa are widely described in the literature [6], SNB-related ones are still under investigation and poorly described. Tracer safety profiles are well known given the experience gathered from routinary administration during lymphangiographic procedures [45]. While no severe allergic reactions after tracer injection were reported, Fumadò et al. reported one case of urinary tract infection (UTI) due to intraprostatic injection and another of ureteral infraction requiring surgical repair while performing an SNB outside the standard ePLND template [18]. Neither further complications nor other SNB-related adverse events were mentioned in the examined studies. 

## 4. Discussion

Although still considered experimental, most up-to-date literature outlines new appealing perspectives on the role of SNB for PCa. Regarding the diagnostic outcomes, the high NPV, both “per patient” and “per node”, represents one of the most relevant results. These findings are comparable with the “per patient” analysis presented by Wit et al. in their systematic review [16]. On the other hand, despite a generally high “per patient” sensitivity, the low “per node” sensitivity demonstrated SNB therapeutic inadequacy compared to ePLND. However, the combination of ePLND and SNB might lower the false-negative rate since it has been proven that aberrant SLN drainage occurs in a significant number of patients undergoing SNB [19]. Despite these attractive results, high SNB methodological variability was observed among different experiences regarding technical aspects and diagnostic accuracy evaluation. Moving forward concerning the transperineal approach [46], only a few reports tested the reliability of such a technique in intraprostatic tracer administration. Moreover, the 2019 Consensus Panel Meeting on SNB for PCa indicated that all studies should report at least sensitivity, specificity, NPV, positive predictive value (PPV), FN, and FP rates [47]. According to our findings, sensitivity and NPV were the most investigated variables. Nevertheless, diagnostic accuracy might be more appropriately evaluated by other underreported parameters, such as the negative likelihood ratio (NLR). It represents a well-suited parameter for SNB diagnostic performance evaluation. NLR refers to the ratio between the probability that all the SLNs in a pN1 patient are negative (false-negative) and the probability that all the ICG-stained LNs are negative in a pN0 patient (true-negative). Claps et al. reported an NLR of 8.6% “per patient” and 53.0% “per node” [17]. 

Notably, it should be pointed out that only a minority of the included articles distinguished between a “per node” and “per patient” specific outcomes potentially influencing the SNB diagnostic performance definition. 

As for diagnostic measurements, procedural heterogeneity itself represents a limit in the evaluation of SNB’s role. Previous experiences described multiple approaches with noticeable differences, mostly related to tracer selection and prostate injection site; hence definitive comparative studies are eagerly awaited. Tracers’ selection represents the critical step to define the following surgical planning. To date, the most used tracer in the PCa SNB setting has been the 99mTc-nanocolloid. In 2011 Van der Poel et al. first described its role in SLN for PCa [48]. From a practical standpoint, tracer administration must be planned the day before surgery and carried out through an intraprostatic ultrasound-guided injection. Lymphoscintigraphy is performed on the same day. SLN detection may be eventually enhanced through combining single photon emission computed tomography (SPECT) examination for higher accuracy. Intraoperative SLN detection is performed using an operative gamma probe and comparing the anatomical intraoperative findings with preoperative nuclear medicine imaging. Later, the development of the Drop-In Gamma Probe made it possible to apply radio-guided technology to the robotic platform [49]. Once the SLN location has been detected, targeted dissection can be performed by harnessing radio-guided signals, which allows identifying the target node overcoming the anatomical obstacles represented by surrounding tissues and structures that might conceal it. On the other hand, it must be acknowledged that such procedures require the availability of highly specialized nuclear medicine technology and complex perioperative management while entailing radiation exposure to both patients and operators. To overcome such limitations, ICG has been studied as a safe, cost-effective, and radiation-free alternative. ICG is a water-soluble dye widely used in diagnostic medicine, which can be administered as a non-conjugated molecule. Thanks to its favorable safety profile and non-radioactive near-infrared (NIR) fluorescence properties, it represents a feasible option for performing a radiation-free lymphography and SNB, which was first described by Yuen et al. [30]. In this setting, an intraprostatic tracer injection is usually performed intraoperatively. Subsequent exposition to NIR light allows visualization and harvesting of the SLNs through a targeted dissection. Hybrid markers, namely ICG-99Tc nanocolloids, resulting from the combination of the aforementioned tracers and providing both radioactive and fluorescent guidance to SNB, have been initially described by Kleinjan et al. [31]. As for non-conjugated radiotracers, a preoperative lymphoscintigraphy combined with SPECT/TC is performed to obtain a preprocedural lymphatic roadmap, whereas the combination of both NIR light and gamma-probe acoustic signal guides the SNB procedure intraoperatively. A further, though less widely reported alternative, is represented by magnetic tracers. Initially described for breast cancer, SPION has been used for the first time by Winter et al. in the SNB setting for PCa [36]. Like their counterparts, magnetic tracers are transrectally injected under ultrasound guidance the day before surgery, while magnetic activity is intraoperatively detected using a magnetometer. More recently, PSMA radio-guided surgery (RGS) has gained a growing interest as an innovative technique in the field of PCa surgical management. As previously discussed, PSMA radiopharmaceutical compounds are currently deployed in the diagnostical field for both PCa primary staging and longitudinal monitoring (e.g., restaging after BCR). Combining preoperative staging and intraoperative labeling, PSMA RGS aims to outperform the diagnostical yield of ordinary ePLND templates. Originally described by Maurer et al. [50,51], PSMA RGS has gained raising interest in both settings of salvage and primary treatment. Patients undergoing PSMA RGS are scheduled for a preoperative SPECT/TC the day before surgery, and a PSMA-labeled radio-compound is administered intravenously at that time. Afterward, PSMA-avid tissues are detected intraoperatively using gamma probes without requiring intraprostatic tracer administration. 

Given the wide range of available compounds, comparative studies have been conducted on this topic. Mazzone et al. compared the diagnostic and oncological outcomes of ICG-SNB and “hybrid”-SNB to highlight the superiority of the “hybrid”-SNB technique in both areas. Nonetheless, their analysis was affected by some limitations due to the relatively small dimension of the ICG group and the short oncological follow-up [52]. In a recently published phase-II randomized trial, Wit et al. compared a hybrid ICG-99mTc-nanocolloid tracer with “multi-step” sequential administration of preoperative 99TC-nanocolloid and intraoperative free ICG tracers [53]. The main outcomes were considered the number of LNs identified by fluorescence guidance and the PPV for positive LNs detection. Although the number of fluorescent LNs removed was higher for the sequential approach, the hybrid group reported a higher rate of tumor-positive fluorescent LNs. Therefore, the authors concluded that hybrid ICG-99mTc-nanocolloid outperforms sequential tracer administration. Engels et al. presented a retrospective analysis comparing radio-guided SNB and magnetic SNB techniques. They described a comparable detection rate (98.18% vs. 98.11%, respectively) [54]. Despite the feasibility of the magnetometer-guided technique, this group underlined some contraindications of the magnetic-guided technique in patients bearing pacemakers or other implanted electronic devices, in cases of iron hypersensitivity as well as in hemochromatosis or other iron overload conditions [54]. Moreover, it should be taken into account that the presence of any metal implant or prosthesis may negatively affect the diagnostic yield of the technique. Hence, the authors concluded that radioisotope-guided techniques might be more suitable in such cases. 

Despite the aforementioned findings, developing a standardized PCa SNB procedure is still burdened by the lack of extensive randomized comparative studies evaluating its cost-effectiveness. Preliminary evidence suggests that the hybrid ICG-99mTc-nanocolloid tracer outperforms other compounds in diagnostical and oncological outcomes [52,53]. However, it should be considered that radiotracer administration pathways for SNB and PSMA RGS require precisely scheduled preoperative planning and close cooperation with a highly specialized nuclear medicine department. On the other hand, the ICG-free SNB technique represents a more feasible option as the main advantages lie in the absence of radiation exposure for patients and medical staff, as well as in intraoperative-only administration. Furthermore, an ICG-free setting allows easier logistical preoperative planning since no preoperative SPECT/CT is required. 

PSMA RGS represents an innovative therapeutic frontier for PCa staging. This technique has been proposed previously as an option for salvage surgical treatments [50,51,55,56]. In this setting, PSMA RGS showed promising results: in the largest cohort of 121 patients who underwent radio-guided salvage surgery after BCR, Horn et al. reported a 66% rate of complete biochemical response [57]. Recently de Barros et al. published the first prospective feasibility study on PSMA RGS in patients with recurrent PCa. The procedure was technically feasible for intraoperative detection of both nodal and local PCa recurrences [58]. Regarding the primary setting, Gandaglia et al. presented interim analyses of a phase 2 study aimed at describing PSMA-RGS during robot-assisted radical prostatectomy (RARP) for cN0M0 patients. Preliminary results showed that this methodology was safe and feasible, with acceptable specificity but an optimizable sensitivity for micrometastases detection [20]. Despite its attractiveness, preliminary data do not highlight any advantages for PSMA RGS over conventional SNB techniques. Thus, further experiences based on larger cohorts are awaited to draw definitive conclusions. 

Considering the tracer deposition, it is still debatable whether to perform an intraprostatic or intralesional tracer injection. PCa is by nature a multifocal neoplasm, arising most frequently in the peripheral zone. Based on these pathological cornerstones, as seen in Table 1, tracer injection was mostly performed covering a whole but peripheral-only template. Recently, an intralesional template has been investigated. Wit et al. recently published a randomized phase II trial comparing intralesional and traditional intraprostatic templates regarding positive LNs detection in the final pathology report [59]. Hybrid ICG-99TC nanocolloids tracer was administrated in two depots of 1 mL in the intralesional administration group or in four depots of 0.5 mL in the peripheral intraprostatic administration group, followed by preoperative lymphoscintigraphy and SPECT/CT. In the intralesional administration group, a significantly higher LNMs detection rate was reported. On the other hand, intralesional administration failed to identify positive LNs from non-index prostatic lesions. Moreover, intralesional injection-guided SNB did not improve metastases-free survival at 4–7 years. Therefore, the authors concluded that the appropriate tracer administration strategy should combine intralesional and intraprostatic approaches [59]. 

As mentioned above, some studies investigated the impact of SNB on survival outcomes. Preliminary data based on short to intermediate follow-up have shown a limited improvement [19,22,24,44,52,60]. However, these results should be interpreted with caution due to the lack of long-term data and a randomized design, as well as the high methodological heterogeneity. Moreover, the prognostic role of positive LNs removal outside the standard template is yet to be determined. Nevertheless, some promising results have already been published: Mazzone et al. reported “hybrid” tracer SNB to be associated with a 60-month BCR rate improvement for patients undergoing RP with ePLND [52]. Consistent with previous results, Grivas et al. outlined an improved 5-year BCR in the SNB cohort for patients undergoing RARP with ePLND [61]. Finally, Claps et al. assessed the prognostic significance of additional ICG-guided LND for patients undergoing RARP combined with ePLND. Patients were equally matched in two groups according to their risk group category. The control group received a standard ePLND, while the second cohort underwent ICG-guided LND followed by ePLND. Patients included in ICG plus ePLND group showed a significant BCR-free survival improvement compared to standard ePLND, particularly when considering the pN-positive cohort [44]. 

Regarding safety, our review found that SNB procedures were safe with a limited number of related adverse events. The authors reported a low number of adverse events directly associated with SNB and no major procedure-related complications. In this regard, according to Mazzone et al. SNB was not burdened by complications classified as above Clavien-Dindo grade IIIa in a large patient cohort [52]. Nonetheless, it should be considered that SLN dissection outside the usual ePLND template may be technically challenging and expose patients to the risk of atypical complications [18]. 

Overall, in the context of newly diagnosed clinically localized cN0 PCa, adopting SNB would be appropriate. This technique benefits candidates for ePLND who carry a low probability of harboring LNMs. The high median NPV further supports the notion that, as a general principle, if LNs are negative, the likelihood of finding positive nodes at final pathology is significantly low. 

Our review is burdened by several limitations. An important one is represented by the lack of data regarding “per node” assessment. This drawback hampers the overall evaluation of SNB diagnostical performance. The lack of long-term outcomes considering oncological outcomes for SNB is of concern for an effective survival estimation. It must be further pointed out that the most important drawback of SNB procedures in PCa is the lack of reliable techniques for intraoperative SN analysis. Here, Winter et al. reported promising preclinical results about one-step nucleic acid amplification (OSNA) quantifying the CK19-mRNA copies [62]. A further limitation is represented by the poorly investigated impact of histological subtypes in this specific scenario. Histological subtypes have been recognized as markers of biological aggressiveness and prognostic factors in several urological malignancies, including PCa [63,64]. Notwithstanding, to our knowledge this is the first review including a “per node” evaluation of SNB techniques for PCa.

## 5. Conclusions and Future Prospectives

The diagnostic and therapeutic role of SNB in PCa is yet to be fully determined. At the same time, the currently widespread of innovative preoperative staging tools prospects innovative scenarios in PCa management. In this regard, Hinsenveld et al. described how PSMA PET/CT and SNB could be complementary in the nodal staging [24]. All patients enrolled underwent preoperative PSMA PET/CT and ePLND, while SNB was performed only in PET/CT negative ones. Results showed that combining both modalities led to a 94% accuracy for nodal staging and that combined sensitivity was 100% since all node-positive patients were correctly staged. Furthermore, applying such a technique could also determine some influences in the field of radiation therapy. According to the recently published results by de Barros et al., SNB can be performed to stratify patients’ risk to identify those with positive SLNs who are likely to benefit from an adjuvant radiation template, including pelvic nodes, and those who could safely undergo prostate-only treatment [65]. Recently, Berrens et al. described a bimodal SLN hybrid-tracer procedure based on multi-fluorescence imaging [66]. The multi-fluorescence radio-guided roadmap that can be obtained allows distinguishing prostatic lymphatic drainage from the drainage system of the adjacent organs, thus improving SNB selectivity. In conclusion, SNB represents a promising perspective that might bring about a shift from risk-based to target-based surgery in the landscape of PCa management. Further, well-designed studies are pending to draw solid conclusions regarding this topic.

## Figures and Tables

**Table 1 diagnostics-13-02543-t001:** Diagnostic test accuracy outcomes among the studies evaluating SNB techniques.

Study	Patient		Procedure		LNS, Mean (n.)		Per Patient (n.)				Per Node (n.)		
	n.	EAU Risk Group IR/HR-LA	Tracer	Injection Site	ePLND	SLNs	Se. (%)	NPV (%)	FN (%)	FP (%)	Sp. (%)	Se. (%)	NPV (%)
Claps (2022) [17]	219	149/70	ICG	TZ bilaterally	22.0	6.0	91.4	97.0	NR	NR	NR	63.5	98.1
Fumadò (2022) [18]	64	NR	99mTc-NC	Intralesional/peripheral to the region of interest	17.3	3.3	94.4	97.8	5.5	4.3	NR	67.3	98.8
Lannes (2022) [19]	162	106/56	99mTc-NC	PZ bilaterally	16.0	4.9	95.4	99.2	4.5	0.7	NR	68.9	98.2
Gandaglia (2022) [20]	12	4/8	99mTc-PSMA-I&S	Intravenous administration	22.0	NR	67.0	90.0	NR	NR	99.0	50.0	96.0
Gondoputro (2022) [21]	12	0/12	99mTc-PSMA-I&S	Intravenous administration	17.0	NR	NR	NR	NR	NR	96.0	76.0	91.0
Shimbo (2020) [22]	100	60/40	ICG	Prostate base and apex bilaterally	24.5	8.8	NR	NR	NR	NR	64.8	34.1	98.2
Meershoek (2020) [23]	15	NR	Hybrid ICG-Tc	Intraprostatic	NR	3.8	76.9	99.0	NR	NR	98.8	NR	NR
Meershoek (2020) [23]	10	NR	99mTc-NC + ICG	Intraprostatic	NR	5.6	NR	NR	NR	NR	99.3	NR	NR
Hinsenveld (2019) [24]	53	8/45	Hybrid ICG-Tc, Tc+ICG	PZ bilaterally	22.0	7.0	67.0	NR	NR	NR	94.0	100.0	NR
Geißen (2019) [25]	218	97/121	SPION	PZ bilaterally	NR	NR	88.1	93.9	4.1	0.9	NR	NR	NR
Winter (2018) [26]	50	NR	SPION	Intraprostatic	16.5	9.0	85.7	94.9	14.3	2.8	NR	NR	NR
Stanik (2018) [27]	20	10/10	SPION	PZ bilaterally	20.0	5.0	80.0	94.0	20.0	0.0	96.0	56.0	91.0
Miki (2016) [28]	28	NR	ICG	NR	NR	4.9	100.0	100.0	0.0	0.0	NR	NR	NR
Kjolhede (2015) [29]	83	NR	99mTc-NC	Peri-lesional	19.0	2.5	95.2	98.0	4.8	3.9	NR	NR	NR
Yuen (2015) [30]	66	NR	ICG	Intraprostatic	22.0	4.0	100.0	100.0	0.0	0.0	NR	NR	NR
Kleinjan (2014) [31]	40	NR	ICG-99mTc-NC	Intraprostatic	12.0	4.0	75.0	93.8	25.0	0.0	NR	NR	NR
Manny (2014) [32]	50	NR	ICG	Intraprostatic	14.0	NR	100.0	100.0	0.0	0.0	NR	NR	NR
Muck (2014) [33]	819	NR	99mTc-NC	Intraprostatic	11.0	3.7	NR	97.1	NR	NR	NR	NR	NR
Rousseau (2014) [34]	203	NR	99mTc-sulfur	Intraprostatic	34.0	5.6	91.4	98.2	8.6	0.0	NR	NR	NR
Stanik (2014) [35]	32	NR	99mTc-NC	Intraprostatic	17.0	4.0	89.3	93.9	10.7	8.0	NR	NR	NR
Winter (2014) [36]	20	NR	SPION	Intraprostatic	24.0	7.0	100.0	100.0	0.0	0.0	NR	NR	NR
Ponholzer (2012) [37]	54	NR	99mTc-NC	Intraprostatic: 3 locations in each lobe	16.0	2.1	93.3	97.4	6.7	0.0	NR	NR	NR
Hinev (2009) [38]	26	NR	99mTc-NC	Intraprostatic: 4 locations in PZ	13.0	3.0	81.8	88.2	18.2	0.0	NR	NR	NR
Brenot-Rossi (2008) [39]	100	NR	99mTc-NC	Intraprostatic	7.0	3.0	100.0	100.0	0.0	0.0	NR	NR	NR
Meinhardt (2008) [40]	35	NR	99mTc-NC	Intraprostatic	13.0	NR	100.0	100.0	0.0	6.3	NR	NR	NR
Weckermann (2007) [41]	33	NR	99mTc-NC	Intraprostatic bilaterally	18.0	7.0	97.6	98.5	2.4	8.3	NR	NR	NR
Fukuda (2007) [42]	42	NR	99mTc-phytate	Intraprostatic	26.0	NR	91.7	96.7	8.3	3.3	NR	NR	NR
Hacker (2006) [43]	20	NR	99mTc-HAS	Intraprostatic bilaterally in 3 sites	14.0	NR	100.0	100.0	0.0	0.0	NR	NR	NR

Abbreviations are as follows: ePLND = extended pelvic lymph nodes dissection; IR = intermediate risk; HR = high risk; LA = locally advanced; LN = lymph node; SLN = sentinel lymph node; Se. = sensibility; Sp. = specificity; NPV = negative predictive value; FN = false negative; FP = false positive; TZ = transition zone; PZ = peripheral zone; ICG = indocyanine green; 99mTc-NC= technetium 99m-nanocolloids; 99mTc-PSMA-I&S = technetium 99m prostate-specific membrane antiger for imaging and surgery; SPION = superparamagnetic iron oxide nanoparticles; HSA = human serum albumin, NR = not reported. n. = stands for number of patients.

**Table 2 diagnostics-13-02543-t002:** Median values for standard diagnostic variables.

Variable	Setting	Papers (n.)	Min. Value (%)	Max. Value (%)	Median (%)	IQR (%)
Sensitivity	Per patient Per node	24 8	67.0 34.1	100.0 100.0	92.5 65.4	82.8–100.0 51.5–74.2
NPV	Per patient Per node	24 7	88.2 91.0	100.0 98.8	97.9 98.1	94.2–100.0 91.0–98.2
Specificity	Per node	7	64.8	99.3	96.0	94.0–99.0
FN	Per patient	20	0.0	25.0	4.7	0.0–10.2
FP	Per patient	20	0.0	8.3	0.0	0.0–3.8

Abbreviations are as follows: IQR = interquartile range, NPV = negative predictive value, FN = False Negative, FP = false positive. n. = stands for number of patients.

**Table 3 diagnostics-13-02543-t003:** Oncological outcomes among the studies evaluating SNB techniques.

Study	Follow Up		Outcomes			
	Median (mos)	IQR	PSA Persistency (%)	BCR (%)	Clinical Progression (%)	RFS, Mean (mos)
Claps (2022) [44]	37.0	24.9–48.8	NR	24.2 vs. 34.1 (overall cohort) 45.9 vs. 75.1 (pN+ cohort)	NR	NR
Fumadò (2022) [18]	32.2	NR	9.4	22.4	10.3	NR
Lannes (2022) [19]	12.0	NR	NR	14.2	NR	NR
Gandaglia (2022) [20]	1.0	NR	25.0	NR	NR	NR
Gondoputro (2022) [21]	13.0	NR	41.7	16.7	NR	NR
Meershoek (2020) [23]	23.3	7.0–57.0	NR	16.0	NR	NR
Yuen (2015) [30]	16.5	6.0–27.0	3.0	3.0	NR	NR
Kleinjan (2014) [31]	10.5	3.0–35.0	NR	37.5	NR	NR
Muck (2014) [33]	60.0	34.8–82.8	NR	38.6	NR	56.4

Abbreviations are as follows: mos = months; SNB = sentinel node biopsy; IQR = interquartile range; BCR = biochemical recurrence; RFS = recurrence-free survival; NR = not reported; pN = pathological node.

**Table 4 diagnostics-13-02543-t004:** Median follow-up among SBN studies evaluating oncological outcomes.

Variable	Papers (n.)	Min. Value	Max. Value	Median	IQR
Follow-up (mos)	9	1.0	60.0	16.5	11.3–34.6
PSA persistency (%)	4	3.0	41.7	17.2	4.6–37.5
BCR (%)	8	3.0	38.6	16.7	14.0–37.5

Abbreviations are as follows: mos = months; IQR = interquartile range; BCR = biochemical recurrence. n. = stands for number of patients.

## Data Availability

Not applicable.

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
