# Peer review of "Sentinel Lymph Node Biopsy in Prostate Cancer: An Overview of Diagnostic Performance, Oncological Outcomes, Safety, and Feasibility"

_diagnostics, 2023, doi:10.3390/diagnostics13152543_

Round 1
Reviewer 1 Report
The work presented by Rossin et al. is of great interest as it explores the practice of sentinel lymph node detection in prostate cancer, which despite being underutilized, has shown to be an innovative and effective method. The paper is well written, clear, and flows smoothly. I only have a few minor suggestions for improvement.
Firstly, although the authors mentioned that this is not a systematic review, it would be beneficial if they could provide the total number of search results obtained and the number of excluded papers that did not meet the inclusion criteria.
Secondly, it would be valuable if the authors included a brief commentary expressing their opinion on the most appropriate setting for implementing this practice. For instance, they could discuss whether it is best utilized during primary surgery for staging purposes or before radiotherapy to aid in determining the radiation field. This discussion would help the readers gain insights into the practical applications of the method.
Lastly, it would be interesting to know if the analyzed papers addressed the frequency of tracer-positive sentinel nodes being found outside the usual template for lymphadenectomy. If such instances were reported, it would be informative to understand whether these nodes were surgically removed in those cases.
Overall, I find the work by Rossin et al. to be valuable and well-presented.
Author Response
Reviewer #1
The work presented by Rossin et al. is of great interest as it explores the practice of sentinel lymph node detection in prostate cancer, which despite being underutilized, has shown to be an innovative and effective method. The paper is well written, clear, and flows smoothly. I only have a few minor suggestions for improvement.
We thank the Reviewer for her/his encouraging words, we hope that our revisions will improve the manuscript.
Firstly, although the authors mentioned that this is not a systematic review, it would be beneficial if they could provide the total number of search results obtained and the number of excluded papers that did not meet the inclusion criteria.
We thank the Reviewer for this comment. We clarified this point in the Methods’ paragraph (see lines 84-86).
Secondly, it would be valuable if the authors included a brief commentary expressing their opinion on the most appropriate setting for implementing this practice. For instance, they could discuss whether it is best utilized during primary surgery for staging purposes or before radiotherapy to aid in determining the radiation field. This discussion would help the readers gain insights into the practical applications of the method.
We thank the Reviewer for this remarkable comment. We updated the Discussion according to his/her suggestion based on our review and experience (see lines 342-347).
Lastly, it would be interesting to know if the analyzed papers addressed the frequency of tracer-positive sentinel nodes being found outside the usual template for lymphadenectomy. If such instances were reported, it would be informative to understand whether these nodes were surgically removed in those cases.
We thank the Reviewer for this hint. However, we considered only reports adopting the extended templates’ limits as positive nodes outside such boundaries are considered metastases by definition.
Overall, I find the work by Rossin et al. to be valuable and well-presented.
We thank again the Reviewer for for her/his appreciation.
Reviewer 2 Report
Good work
Author Response
Reviewer #2
Good work.
We thank the Reviewer for his/her kind words, we hope that our revisions will improve the manuscript.
Reviewer 3 Report
The manuscript entitled “Sentinel lymph node biopsy in prostate cancer: an overview on diagnostic performance, oncological outcomes, safety and feasibility” by Rossin et al. aim to review the most recent literature regarding the SLN in PCa, considering the lack of solid evidence and standardized evidence. Although the interesting topic, the manuscript has two main flaws which are the hybrid review (narrative? Systematic?) and the lack of a proper results section. Nevertheless, considering the topic and the data retrieved, these flaws could be resolved properly with a thorough review of the manuscript. Further suggestion are reported followingly:
INTRODUCTION
36-45: About epidemiological data on PCa, also see DOI: 10.2144/fsoa-2020-0210
53-55: I would be more cautious regarding this affirmation considering the capabilities, not fully explored, of PET PSMA. To this regard please see doi: 10.3390/diagnostics12112594
76-122: all this part could be moved to the discussion. In the introduction, a brief description of SLN in PCa and its limitations is enough. It would be better to link this issue with the aim of your study.
MATERIALS AND METHODS
128-136: is this a narrative o systematic review? In the first case, this section could be more flexible considering the lack of a proper methodology for retrieving data and studies. In the second case, you should be more meticulous in reporting the inclusion and exclusion criteria as well as the string of research utilized and, most importantly, a prisma flow diagram reporting the studies retrieved. This is a major issue.
RESULTS
137: Albeit the tables are quite self-explanatory, a brief summarization of the main findings of the study analyzed should be reported. The same issue is valid for the following subparagraph. In synthesis, you should not place a table and say to the reader “read it yourself”.
201: the issue regarding safety profiles should be further discussed according to the findings retrieved from the studies analyzed.
DISCUSSION
217-240: avoid redundancy within the paragraph and with the introduction.
246-268: as before.
331: about the limitations, the lack of statistical support is not a limitation itself considering that your study is a narrative/systematic review (define your type of review). I would suggest you to analyze and focus on why the SLN has not been widely used in clinical practice
minor typos
Author Response
Reviewer #3
The manuscript entitled “Sentinel lymph node biopsy in prostate cancer: an overview on diagnostic performance, oncological outcomes, safety and feasibility” by Rossin et al. aim to review the most recent literature regarding the SLN in PCa, considering the lack of solid evidence and standardized evidence. Although the interesting topic, the manuscript has two main flaws which are the hybrid review (narrative? Systematic?) and the lack of a proper results section. Nevertheless, considering the topic and the data retrieved, these flaws could be resolved properly with a thorough review of the manuscript. Further suggestion are reported followingly:
INTRODUCTION
36-45: About epidemiological data on PCa, also see DOI: 10.2144/fsoa-2020-0210
We thank the Reviewer for this suggestion. Accordingly, we updated the reference list with this further manuscript (see line 38).
53-55: I would be more cautious regarding this affirmation considering the capabilities, not fully explored, of PET PSMA. To this regard please see doi: 10.3390/diagnostics12112594
We thank the Reviewer for this comment. Accordingly, we added the suggested reference and updated the manuscript (see lines 53-55).
76-122: all this part could be moved to the discussion. In the introduction, a brief description of SLN in PCa and its limitations is enough. It would be better to link this issue with the aim of your study.
We thank the Reviewer for this suggestion. We updated the manuscript accordingly (see lines 211-256).
MATERIALS AND METHODS
128-136: is this a narrative o systematic review? In the first case, this section could be more flexible considering the lack of a proper methodology for retrieving data and studies. In the second case, you should be more meticulous in reporting the inclusion and exclusion criteria as well as the string of research utilized and, most importantly, a prisma flow diagram reporting the studies retrieved. This is a major issue.
We thank the reviewer for this comment. We specified the narrative design of this review in the abstract, in the end of the Introduction (see lines 78-80 as well as in the Methods’ section (see line 86).
RESULTS
137: Albeit the tables are quite self-explanatory, a brief summarization of the main findings of the study analyzed should be reported. The same issue is valid for the following subparagraph. In synthesis, you should not place a table and say to the reader “read it yourself”.
We thank the reviewer for this comment. Accordingly, we further expanded each Results’ section paragraphs (see lines 107-118 and 160-161).
201: the issue regarding safety profiles should be further discussed according to the findings retrieved from the studies analyzed.
We thank the reviewer for this comment. As we stressed, the very low rate of adverse events makes SNB a safe procedure. We specified this point in the Discussion (see lines 335-341).
DISCUSSION
217-240: avoid redundancy within the paragraph and with the introduction.
We thank the reviewer for this valuable comment. Accordingly, we updated the manuscript (lines 183-198).
246-268: as before.
We thank the reviewer for these comments. Accordingly, we updated the manuscript (lines 257-271).
331: about the limitations, the lack of statistical support is not a limitation itself considering that your study is a narrative/systematic review (define your type of review). I would suggest you to analyze and focus on why the SLN has not been widely used in clinical practice.
We thank the reviewer for this important suggestion. Accordingly, we highlighted these aspects in the Discussion section (see lines 348-355).
Round 2
Reviewer 3 Report
The authors improved the manuscript accordingly to previous suggestions. No further corrections are required.
nothing of relevant
Author Response
Thank you for the time you spent revising our manuscript.